# A Satellite-Free Centromere in *Equus przewalskii* Chromosome 10

**DOI:** 10.3390/ijms24044134

**Published:** 2023-02-18

**Authors:** Francesca M. Piras, Eleonora Cappelletti, Wasma A. Abdelgadir, Giulio Salamon, Simone Vignati, Marco Santagostino, Lorenzo Sola, Solomon G. Nergadze, Elena Giulotto

**Affiliations:** 1Department of Biology and Biotechnology “Lazzaro Spallanzani”, University of Pavia, 27100 Pavia, Italy; 2Oasi di Sant’Alessio, Sant’Alessio con Vialone, 27016 Pavia, Italy; 3Independent Researcher, 27100 Pavia, Italy

**Keywords:** *Equus przewalskii*, karyotype evolution, centromere, satellite DNA, ChIP-seq

## Abstract

In mammals, centromeres are epigenetically specified by the histone H3 variant CENP-A and are typically associated with satellite DNA. We previously described the first example of a natural satellite-free centromere on *Equus caballus* chromosome 11 (ECA11) and, subsequently, on several chromosomes in other species of the genus *Equus*. We discovered that these satellite-free neocentromeres arose recently during evolution through centromere repositioning and/or chromosomal fusion, after inactivation of the ancestral centromere, where, in many cases, blocks of satellite sequences were maintained. Here, we investigated by FISH the chromosomal distribution of satellite DNA families in *Equus przewalskii* (EPR), demonstrating a good degree of conservation of the localization of the major horse satellite families 37cen and 2PI with the domestic horse. Moreover, we demonstrated, by ChIP-seq, that 37cen is the satellite bound by CENP-A and that the centromere of EPR10, the ortholog of ECA11, is devoid of satellite sequences. Our results confirm that these two species are closely related and that the event of centromere repositioning which gave rise to EPR10/ECA11 centromeres occurred in the common ancestor, before the separation of the two horse lineages.

## 1. Introduction

The centromere is the site of kinetochore assembly required for chromosome segregation during cell division. The centromeric function is epigenetically specified by the histone H3 variant CENP-A, which is highly conserved during evolution [1]. On the other hand, the underlying DNA sequences are extremely variable and are neither necessary nor sufficient to determine the centromeric function [2,3,4,5]. In mammals, centromeres are typically characterized by extended arrays of tandemly repeated sequences, called satellite DNA. These sequences represent the most rapidly evolving DNA sequences in eukaryotic genomes [6] and their presence has so far hampered comprehensive molecular analysis of these intriguing loci. In this context, our discovery that the species of the genus *Equus* are characterized by an extraordinarily high number of centromeres completely devoid of satellite DNA made these species a good model system for studying the epigenetic control and evolution of the centromere function [7,8,9,10,11,12,13,14,15,16,17].

The exceptional genome plasticity of equid species is reflected in their rapid karyotypic evolution and is marked by numerous events of centromere repositioning and chromosomal rearrangements [18,19,20]. Centromere repositioning, which is the shift of the centromere position on the chromosome without sequence rearrangement [21,22], gave rise to evolutionarily new centromeres [11,18,19,20,23]. A first cytogenetic approach revealed that this phenomenon played a key role during the evolution of equids [11,16,23]. Horse satellite repeats were cloned and analyzed by Southern blotting and FISH in early studies by different groups [24,25,26,27]. We then studied the chromosomal distribution of satellite sequences, namely 37cen and 2PI, in *E. caballus* (domestic horse), *E. asinus* (donkey), *E. grevyi* (Grevy’s zebra), and *E. burchelli* (Burchell’s zebra) which revealed a peculiar uncoupling between centromeric function and satellite repeats. In particular, several centromeres were found to be devoid of satellite DNA whereas blocks of satellite DNA were observed at some chromosomal termini, representing relics of ancestral inactivated centromeres [13,16].

Later on, using an anti-CENP-A antibody in ChIP experiments on chromatin extracted from primary fibroblast cell lines, we proved at the molecular level that several equid centromeres are completely devoid of satellite repeats [7,9,12,13,14,17]. While only one satellite-free centromere was found in the horse, on chromosome 11 (ECA11) [12], 16 were described in the donkey [7], 15 in the Burchell’s zebra, and 13 in the Grevy’s zebra [14]. Interestingly, it was observed that these CENP-A binding domains can slide within a genomic region of a few hundred kbs, probably limited by epigenetic boundaries. These domains were defined as epialleles and the phenomenon was called centromere sliding [7,9,13,14]. Studies on families composed of donkeys, horses, and their hybrid offspring (mule/hinny) revealed that the epialleles are inherited as Mendelian traits but their position can slide in one generation [7] while it is stable during propagation in cell culture, suggesting that the sliding can presumably take place during meiosis [7].

Among equid species, horses are considered the closest extant species to the equid ancestor, whose karyotype was mainly characterized by acrocentric chromosomes [19]. While the centromeres of the domestic horse were deeply studied both at the cytogenetic and at the molecular level [7,8,9,12,15,16,28,29], the centromeres of the other extant horse species, the Przewalski’s horse, are still lacking a detailed characterization.

Przewalski’s horses have never been successfully domesticated. It was suggested that the Przewalski’s horse species is a direct ancestor of the domestic horse [30,31]. However, this hypothesis is controversial with data indicating that Przewalski’s and domestic horses may represent two lineages that diverged between 38,000 and 72,000 years ago [32,33,34,35]. Recently, Gaunitz and colleagues demonstrated that *Equus przewalskii* is the feral descendant of horses first herded in the Central Asian steppes about 5500 years ago and their feralization likely involved multiple biological changes such as pigmentation as well as morpho-anatomical characteristics [36,37]. After the discovery of this species in the 1870s in the Asian steppes, Przewalski’s horses become extinct in the wild by 1969 but survived in captivity. Since the 1970s, horses were exchanged between zoos in order to reduce the level of inbreeding [38,39,40].

In the 1990s, reintroduction projects brought the species back in the wild but the current population is still endangered and entirely descending from only 13 founding individuals [32,41]. At present, the worldwide population of the living Przewalski’s horses has reached nearly 2500 individuals [42]. Among them, approximately 1360 horses live in the wild in China and Mongolia, 900 are distributed in zoos in Europe, and 120 in wildlife parks in the US [39,40,42,43,44].

Sequence analysis of the control regions from mitochondrial DNA allowed the identification of only four Przewalski’s horse matrilines [41,45]. In 2011, four complete mitogenomes, representing all four surviving mitochondrial lineages, were sequenced and assigned to haplogroups I, II, and III [32,46]. Later on, Achilli and colleagues grouped haplogroups I and II, which were similar, into haplogroup F, which is specific of Przewalski’s horse [47], while haplogroup III resulted closely related to the J and K haplogroups of domestic horses. This observation suggests that there was gene flow among ancestors of the present-day populations or common ancestry [31,47].

The karyotypes of Przewalski’s and domestic horses are very similar with 66 and 64 chromosomes, respectively. The two acrocentric chromosomes, 23 and 24, possibly representing the ancestral configuration, underwent a centric fusion to generate the metacentric chromosome 5 of the domestic horse [48,49,50]. Despite their different chromosomal numbers, the domestic and the Przewalski’s horses can produce viable and fertile offspring [51] and this is the reason why hybrids may be present in Przewalski’s populations both in the wild and in captivity.

Here, we investigated the chromosomal distribution of satellite DNA families in *Equus przewalskii* (EPR). Moreover, we demonstrated at the cytogenetic and the molecular level that the centromere of EPR10 is devoid of satellite DNA sequences. This centromere is located in the genomic region orthologous to the satellite-free centromeric region on ECA11.

## 2. Results

### 2.1. Characterization of a Fibroblasts Cell Line from Equus przewalskii

We established a primary fibroblast cell line from a skin biopsy of a 17-year-old male Przewalski’s horse. To confirm that this animal was a Przewalski’s horse (EPR) and not a hybrid with the domestic horse, we first analyzed its karyotype using G-banding obtained by DAPI staining (Figure 1). As expected for this species, the number of chromosomes was 66 with 12 metacentric pairs, 20 acrocentric pairs, including chromosomes 23 and 24 which are typical of this species, and the XY sex chromosomes. To better identify the EPR chromosomes 23 and 24, which are orthologous to ECA5p and ECA5q [49,50], we performed FISH experiments using the BAC probes CHORI241-17G8 and CHORI241-25H7 specific for ECA5p and ECA5q, respectively (Figure 2). The probes were used to hybridize metaphase spreads from a domestic horse (panel A), from a *E. caballus*/*E. przewalskii* hybrid (2n = 65) (panel B) and from the *E. przewalskii* cell line (panel C).

In the domestic horse, following hybridization with the CH241-17G8 probe, FISH signals were detected at the proximal region of the p-arm of chromosome 5 (Figure 2A left). As expected, in the hybrid individual, FISH signals were detected on the p-arm of the metacentric chromosome 5, derived from the domestic horse parent, and on the acrocentric Przewalski’s horse chromosome 23 (Figure 2B left). Finally, as expected for a Przewalski’s horse, in our individual, the BAC clone hybridized with two acrocentric chromosomes, which compose the pair of homologous chromosomes EPR23 (Figure 2C left). The BAC clone CH241-25H7 was previously mapped by FISH on the q-terminus of chromosome ECA5 [23]. This localization was confirmed in the present work (Figure 2A right). In the hybrid individual, FISH signals were observed on the metacentric horse-derived chromosome 5 and on the acrocentric chromosome 24 of Przewalski’s horse (Figure 2B right). As expected for a Przewalski’s horse, the BAC clone CH241-25H7 was located on the acrocentric EPR24 chromosome pair (Figure 2C right).

To further prove that the fibroblast cell line derives from a Przewalski’s horse, we assembled the sequence of its mitochondrial DNA (access number OQ285861) from whole-genome Illumina reads that were used as input in the ChIP-seq experiment described below. The sequence was then aligned with 104 mitogenomes available in GenBank from domestic horses, Przewalski’s horses, and their hybrids [47,52]. The mitogenome of our individual belongs to haplogroup F, which is typical of Przewalski’s horses.

### 2.2. Chromosomal Distribution of Satellite DNA Families in Equus przewalskii

To analyze the localization of the two major horse satellite DNA families in Przewalski’s horse, we carried out FISH experiments using the previously described plasmids containing 37cen and 2PI satellite sequences as probes [16,53]. Representative metaphase spreads hybridized with the two probes are shown in Figure 3 and the chromosomal distribution of the two satellite sequences is reported in Table 1.

Hybridization signals of the 37cen satellite are present at the primary constriction of all chromosomes with the exception of EPR10 (Table 1). Signal intensities are heterogeneous suggesting that the number of repeats is variable among different chromosomes (Figure 3A). The primary constrictions of chromosomes 2, 6, 18, and 29 show 37cen signals only on a subset of the analyzed chromosomes (Table 1), suggesting that the number of repeats at these loci is low.

Similarly to 37cen, the 2PI satellite was detected at the primary constriction of all chromosomes except EPR10 (Table 1) and the intensity of the signals was heterogeneous among different chromosomes, suggesting inter-chromosomal variability in the number of repeats (Figure 3B). The centromeres of chromosomes 1, 4, 11, 29, X, and Y were labeled only in a subset of the analyzed chromosomes (Table 1), suggesting a low number of repeats at these loci.

To test the presence of satellite DNA families other than 37cen and 2PI, we performed a FISH analysis using total genomic DNA as a probe. We previously used this procedure to identify regions containing highly repetitive sequences because of their different hybridization kinetics compared to single-copy sequences [16]. Even with this probe, chromosome 10 is lacking any hybridization signal while all other chromosomes are labeled at their primary constriction. These observations suggest that the centromere of chromosome 10 is the only one devoid of satellite DNA. EPRX, similarly to what was previously observed for the domestic horse [16], showed two hybridization signals, one at the primary constriction and one at an interstitial position of the q arm (Figure 3C and Table 1). In Figure 3D, an example of double-color FISH with 2PI satellite and total genomic DNA as probes is shown, confirming that chromosome 10 is the only one lacking any signal. The Przewalski’s specific acrocentric chromosomes 23 and 24, which are orthologous to horse chromosome 5p and 5q, respectively [49], are labeled at their centromeric end with the three probes.

### 2.3. ChIP-Seq Analysis of Centromeric Domains

To test, at the sequence level, whether the centromere EPR10 is satellite-free, a ChIP-seq experiment with an anti-CENP-A antibody was carried out on the chromatin extracted from the primary skin fibroblasts described above.

Since a chromosomal-level genome assembly for Przewalski’s horse is not available, given the high karyotypic and sequence identity with the domestic horse [35,49,50,54], we mapped the ChIP-seq reads on the horse reference genome. We used the EquCab2.0 reference where the satellite-free centromeric region of horse chromosome 11 is well assembled [7,12]. Using the pipeline that we developed for the identification of satellite-free centromeres in other equids [7,14], a CENP-A binding domain was identified in the region comprised between 27.59 and 27.79 Mb of chromosome 11 in the reference genome (Figure 4A). In the Przewalski’s horse, this region is located on chromosome 10, which is orthologous to horse chromosome 11 (Figure 4A). The regular shape of this peak indicates that the underlying genomic region is highly conserved between Przewalski’s and domestic horse.

We previously demonstrated that, in equids, homologous chromosomes can carry CENP-A binding domains in different positions, which correspond to different epialleles [7,8,9]. To analyze the organization of epialleles on EPR10 of the animal studied here, we used a single nucleotide variant (SNV)-based approach.

We identified heterozygous nucleotide positions within the centromeric domains using the ChIP reads. These positions are listed in Table 2 and marked as bars in Figure 4A. We detected eight heterozygous nucleotide positions which are distributed along the entire length of the peak, suggesting that the CENP-A binding domains on the two homologs are overlapping (Figure 4A, Table 2).

Since satellite-based centromeres contain megabases of unassembled satellite arrays, we could not obtain ChIP-seq peaks from these centromeres which are probably organized likewise the typical mammalian centromeres, as already shown for the domestic horse [28].

To define the satellite DNA family bearing the centromeric function, we aligned the ChIP-seq reads from CENP-A immunoprecipitated and from input chromatin with the consensus sequences of 37cen and 2PI. To quantify the enrichment of these sequences in CENP-A bound chromatin, we calculated the ratio between normalized read counts in immunoprecipitated and in input DNA (Figure 4B). As control, we used the ERE-1 retrotransposon, which is interspersed throughout the equid genomes and is not expected to be involved in the centromeric function [28]. A 3-fold enrichment was observed for the 37cen satellite while 2PI and ERE-1 were present at similar levels in the two fractions (Figure 4B). These results demonstrated that, as for the domestic horse [28], 37cen is the main functional centromeric satellite sequence.

## 3. Discussion

Since Przewalski’s and domestic horses can breed and give birth to fertile offspring, the presence of hybrid individuals, both in the wild and in captivity, represents one of the major problems in the conservation programs of Przewalski’s horses. Therefore, following the establishment of a new fibroblast cell line, we first tested whether the donor was a Przewalski’s horse not deriving from a recent hybridization with the domestic horse. Indeed, the karyotype of this individual showed 66 chromosomes comprising the acrocentric chromosome pairs 23 and 24, which are typical of this species, whereas the metacentric chromosome 5, which is typical of the domestic horse, was absent. Furthermore, sequence analysis of its mitogenome revealed that this individual belongs to haplogroup F, which is typical of the Przewalski’s species [47].

In previous work from our laboratory, the chromosomal distribution of the two major equid satellite DNA families, 37cen and 2PI, was investigated by FISH in *E. caballus*, *E. asinus*, *E. grevyi*, and *E. burchelli* [16]. This analysis demonstrated that, in the genus *Equus*, several centromeres are devoid of satellite DNA and that, in agreement with the library hypothesis for satellite DNA evolution [55,56], horses, donkeys, and zebras share a common set of satellite DNA families which underwent expansion or shrinkage in the different species [16]. In the horse, either one or both these satellites are present on all primary constrictions, except the one of chromosome 11, which contains the unique satellite-free centromere of this species [12,16].

Here, we showed that there is a good degree of conservation of satellite DNA in Przewalski’s and in domestic horse. However, some differences in the distribution of satellite DNA families between the two species were observed. For example, 37cen signals were detected at the primary constriction of EPR2 while the ortholog of the domestic horse (ECA2) lacks 37cen signals [16]. Moreover, the primary constrictions of chromosomes 1, 4, 11, and X of *E. przewalskii* are labeled by 2PI signals, while, in the orthologous chromosomes of the domestic horse (1, 4, 12, and X), such signals are undetectable [16]. These observations suggest that, in *E. caballus*, the above-mentioned centromeres may lack these satellite DNA families or carry a small number of repeated units which are undetectable at the FISH resolution level.

The high similarity between Przewalski’s and domestic horse was also confirmed following hybridization of metaphase chromosomes with total genomic DNA. This procedure was used to identify regions containing very abundant tandem repeats due to the different hybridization kinetics of highly reiterated sequences versus single copy DNA [16]. The results of the FISH experiments with the three probes (37cen, 2PI, and genomic DNA) suggest that the number of satellite repeats is variable at different primary constrictions (Figure 3 and Table 1). In addition, the appearance of a non-centromeric hybridization signal on the q arm of chromosome X, visible with the genomic DNA probe only, indicates that satellite repeats other than 37cen and 2PI are located at this site. We previously showed that, also in the domestic horse, the q arm of chromosome X contains satellite repeats other than 37cen and 2PI [16].

Interestingly, both major satellite families, 37cen and 2PI, were detected at the centromeres of the two acrocentric chromosomes EPR23 and EPR24 while the 37cen only is present at the primary constriction of the metacentric domestic horse ortholog (ECA5) [16]. Therefore, at the centromere of ECA5 the 2PI satellite is either absent or undetectable by FISH suggesting that this satellite may have been lost during fusion of two ancestral acrocentric chromosomes corresponding to EPR23 and EPR24. The generation of ECA5, in the domestic horse lineage, from a centric fusion was previously proposed [57] and the loss of satellite repeats as a consequence of Robertsonian fusion was recently demonstrated by us in the zebras [14]. However, since the ECA5 synteny is maintained in all Ceratomorpha and disrupted in all equids with the exception of the domestic horse, the phylogenetic history of this chromosome remains controversial. In particular, an ECA5 ortholog is acrocentric in rhinos, whose karyotypes remained quite stable during evolution and similar to its orthologous element in the hypothetical perissodactyl ancestral karyotype [18,19]. This situation suggests an alternative model: a fission event generated the two acrocentric chromosomes corresponding to EPR23 and EPR24. According to this hypothesis, the submetacentric chromosome of the domestic horse may have lost the 2PI satellite during evolution.

The FISH experiments demonstrated that EPR10, the ortholog of ECA11, is the only Przewalski’s chromosome devoid of satellite DNA signals at its primary constriction. Evidence of a possible lack of satellite repeats on Przewalski’s chromosome 10 was provided in an early pioneering study from Ryder and Hansen in 1979 [48] where a GC rich satellite band was purified by density centrifugation. Radioactively labeled cRNA was then hybridized to metaphase spreads from *E. przewalskii.* ChIP-seq results confirmed at the sequence level that the centromere of EPR10 is completely devoid of satellite DNA, similarly to what is observed for its ECA11 ortholog in the domestic horse. In agreement with our cytogenetic data, no other satellite-free centromere was detected in Przewalski’s horse.

The CENP-A enrichment peak detected on EPR10 lays in the same genomic region in which horse CENP-A binding domains are known to slide [7,9,12]. This result suggests that the event of centromere repositioning which gave rise to this centromere occurred in the common ancestor, before the separation of the two horse lineages, between 38,000 and 72,000 years ago [34]. In several domestic horses, two well separated CENP-A binding domains, corresponding to different epialleles on the two homologs, were detected [8,9]. In the Przewalski’s individual analyzed here, we observed a single Gaussian-like peak indicating that this individual is homozygous for centromere position, in agreement with the high level of inbreeding of this species [35,36,41,58,59,60,61].

We could not localize the CENP-A binding domains from the satellite-based centromeres since they contain megabases of satellite arrays which are not assembled in the reference genomes. However, as in the domestic horse [28] the 37cen satellite family is enriched in the CENP-A immunoprecipitated chromatin, thus bearing the centromeric function. The 2PI satellite is not enriched in the CENP-A bound chromatin and, as in the domestic horse, is preferentially distributed at pericentromeric regions.

The high degree of similarity of satellite DNA in terms of composition and distribution between Przewalski’s and domestic horses is in agreement with the recent divergence of the two lineages [32,36]. Interestingly, the localization of these satellite sequences in Burchell’s and Grevy’s zebras varies in the two lineages in a species-specific way [16]. This difference may be related to the fact that the radiation between the two horses is more recent if compared to the separation of the two zebras (>1MYA) [62].

In conclusion, the results of this work confirm that Przewalski’s and domestic horses are phylogenetically very close, supporting the hypothesis that their radiation from a common ancestor occurred in recent evolutionarily times.

## 4. Materials and Methods

### 4.1. Cell Lines

A primary fibroblast cell line was established from a skin biopsy from a male Przewalski’s horse living in captivity in the “Oasi di Sant’Alessio” zoological park. This animal was obtained from the Berlin Zoo (CITES number: De-BLN04031635 16.03.2004 Germany). The biopsy was taken post-mortem after the horse underwent euthanasia following an incurable fracture. The cell line was obtained from small fragments of dermis using a protocol well-established in our laboratory [7,8,9,11,12,15,16,23,28,29,63,64,65,66,67,68]. The horse primary fibroblast cell line was obtained from the skin of a slaughtered animal. The animal was being processed as part of the normal work of the abattoirs.

The cells were cultured in high-glucose DMEM medium (Euroclone, Milan, Italy) supplemented with 20% fetal bovine serum (Euroclone, Milan, Italy), 2 mM L-glutamine (Merck Life Science, Milan, Italy), 1% penicillin/streptomycin (Merck Life Science, Milan, Italy), and 2% non-essential amino acids (Euroclone, Milan, Italy). The cells were maintained at 37 °C in a humidified atmosphere of 5% CO_2_. Using this procedure, relatively pure cultures of skin fibroblasts are obtained. The fibroblasts show a typical elongated shape and grow aligned and in bundles when confluent. As shown also by other groups [69], the above-mentioned medium supports the growth of fibroblasts whereas other cell populations such as keratinocytes need additional supplements and growth factors or are mitotically less active.

Primary fibroblasts from a privately owned hybrid female individual derived from a cross between a domestic horse and a Przewalski’s horse were previously collected in connection with another project and kindly provided by Ernest Bailey from the University of Kentucky.

### 4.2. Metaphase Spread Preparation and FISH

For metaphase spread preparation, mitoses were mechanically collected by blowing the medium on the dish surface. Chromosome preparations were performed with the standard air-drying procedure and were used for karyotype analysis or fluorescence in situ hybridization (FISH).

Whole genomic DNA from Przewalski’s horse fibroblasts was extracted according to standard procedures. BAC clones (CHOR1241-17G8, coordinates in EquCab2.0: chr5:37,011,456–37,167,308; CHOR1241-25H7 coordinates in EquCab2.0: chr5: 98,031,303–98,224,666) and lambda phage 37cen and 2PI clones [16,53] were extracted from 10 mL bacterial cultures with the Quantum Prep Plasmid miniprep kit (BioRad, Milan, Italy), according to supplier instructions. All probes were labeled by nick translation with Cy3-dUTP (Enzo Life Sciences, Milan, Italy) or Alexa488-dUTP (Life Technologies, Monza, Italy) as previously described [64]. FISH was performed as previously described [68].

Chromosomes were counterstained by DAPI. Digital grayscale images for fluorescence signals were acquired with a fluorescence microscope (Zeiss Axio Scope.A1, Zeiss, Göttingen, Germany) equipped with a cooled CCD camera (Teledyne Photometrics, Birmingham, UK). Pseudocoloring and merging of images were performed using the IPLab 3.5.5 Imaging Software (Scanalytics Inc., Fairfax, VA, USA). Chromosomes were identified by DAPI banding according to the published karyotypes [49,50].

### 4.3. ChIP-Seq

Chromatin from primary fibroblasts was cross-linked with 1% formaldehyde, extracted, and sonicated to obtain DNA fragments ranging from 200 to 800 bp. Immunoprecipitation was performed as previously described [7] using a serum against horse CENP-A protein [8]. Paired-end sequencing was performed through an Illumina NovaSeq 6000 platform by IGA Technology Services (Udine, Italy).

### 4.4. Bioinformatic Analysis of ChIP-Seq Data

The identification of satellite-free centromeres was performed as previously described [14]. Reads were aligned with paired-end mode to the EquCab2.0 horse reference genomes with Bowtie2 aligner (version 2.4.2) using default parameters [70,71]. Normalization of read coverage of the ChIP datasets against the input datasets was performed using bamCompare (deepTools suite, 3.5.0 version) [72] using RPKM (Reads Per Kilobase per Million mapped reads) normalization in subtractive mode. The resulting coverage files were visualized using Integrative Genomics Viewer (IGV) software (2.9.2 version). Peaks were obtained with pyGenomeTracks (3.6 version) [73].

To quantify the number of reads corresponding to each repetitive element, the ChIP-seq reads from CENP-A immunoprecipitated chromatin and from input were aligned with the consensus sequences of 37cen (“SAT_EC”on Repbase), 2PI (“ES22”on Repbase and “AH010654.2” on NCBI Nucleotide), and ERE-1 retrotransposon (“ERE1”on Repbase) [74] using Bowtie2 (2.4.2 version) with single-end mode. RPKM normalized read counts were obtained using SAMtools (1.11 version) [75]. To measure enrichment due to immunoprecipitation with CENP-A, the ratio between normalized read counts in the immunoprecipitated and input samples was calculated.

### 4.5. SNV Analysis

To identify heterozygous single nucleotide positions in the CENP-A binding domain, we first removed duplicates from the ChIP BAM files using SAMtools [75,76]. SNVs in the centromeric region were identified using BCFtools (1.11 version) [76] and filtered for quality higher than 20 to exclude low quality calls, as suggested by the program. The minimum number of reads per position required for the analysis was 4, an arbitrary threshold that we previously used to identify SNVs in *E. asinus* [7]. The required frequency of the least frequent variant was equal or more than 0.3 [7]. We used only SNVs present within single copy sequences. As previously described [7], the presence of heterozygous nucleotide positions indicates the region of overlapping between the CENP-A binding domains of the two homologs.

### 4.6. Mitogenomic Analysis

The mitochondrial DNA sequence of our Przewalski’s horse (accession number OQ285861) was obtained from the Input reads aligned to chrM of the EquCab2.0 reference genome. The obtained sequence was aligned with mitochondrial genomes retrieved in GeneBank from 82 domestic horses (JN398377-JN398457, HQ439467, KT757764), 20 Przewalski’s horses (KT221844, KT221845, KT368742-KT368756, KT757761, HQ439484, JN398402, JN398403) and two Przewalski’s horse/domestic horse hybrids (KT368757 and KT368758). Sequence alignment and phylogenetic analysis was performed using MEGAX [77] as previously described by Kvist and collaborators [52].

## Figures and Tables

**Figure 1 ijms-24-04134-f001:**
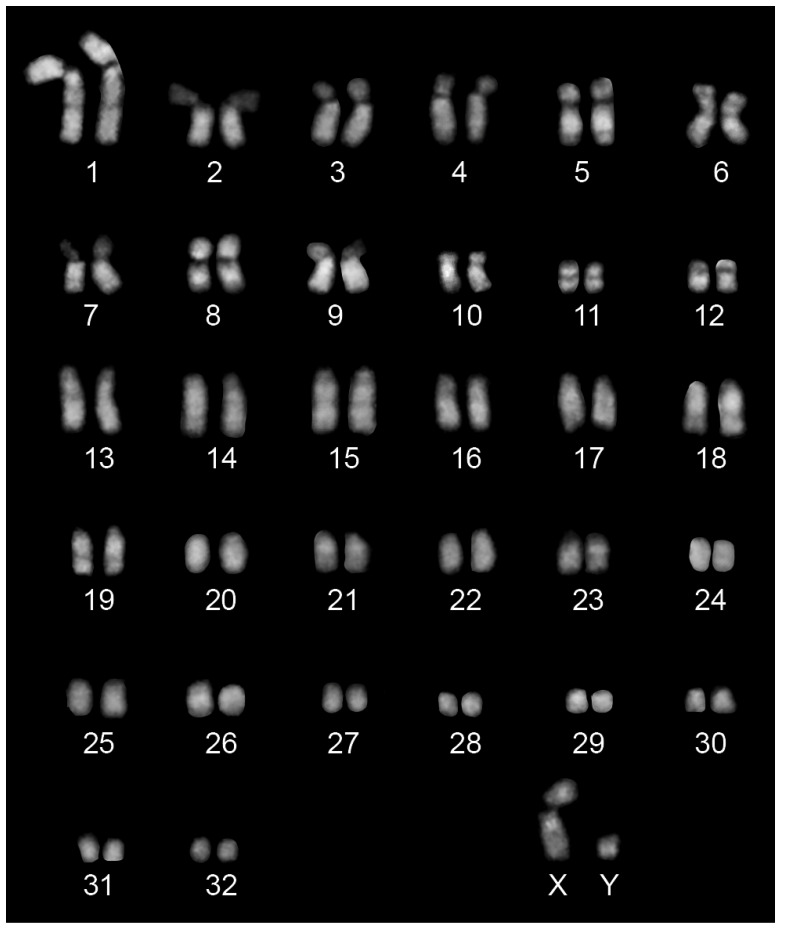
DAPI-banded karyotype of a male *Equus przewalskii*. The karyotype (2n = 66) consists of 24 biarmed autosomes, 40 acrocentric autosomes, a biarmed X chromosome, and the acrocentric Y chromosome. Chromosomes are arranged and numbered according to standard nomenclature.

**Figure 2 ijms-24-04134-f002:**
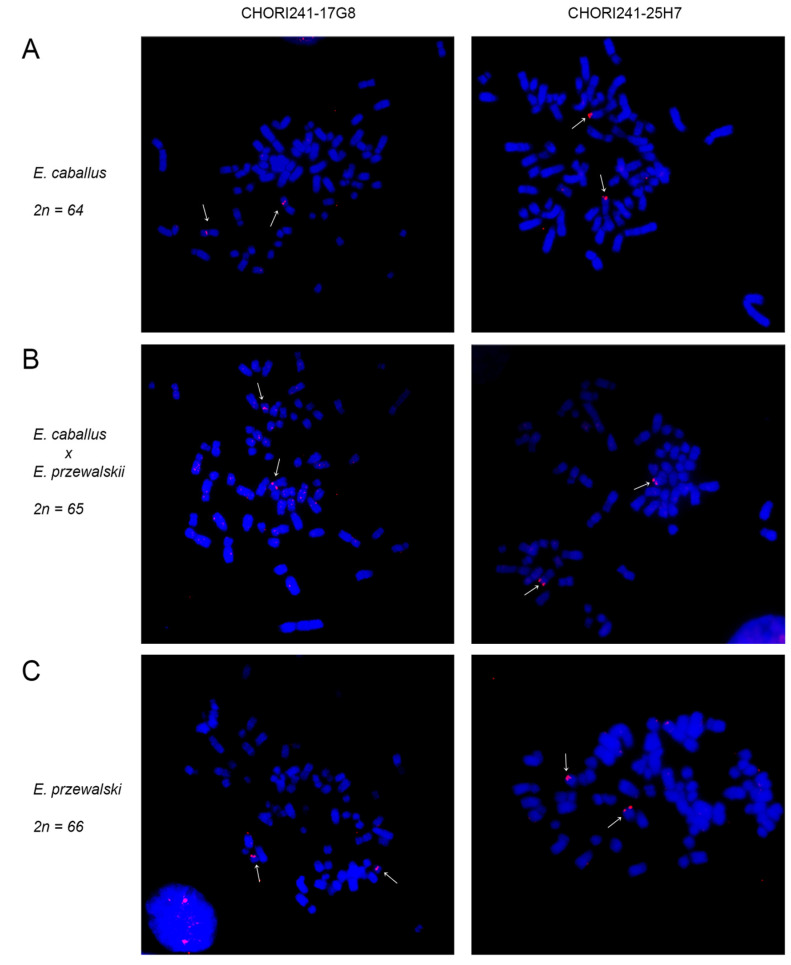
Chromosomal localization of two horse BAC clones specific for ECA5p and ECA5q. Metaphase spreads from *E. caballus* (**A**), *E. caballus* × *E. przewalskii* (**B**), and *E. przewalskii* (**C**) hybridized with BAC clone CHORI241-17G8 (ECA5p) on the left and BAC clone CHORI241-25H7 (ECA5q) on the right. Chromosomes were stained with DAPI (blue). The white arrows indicate BAC signals (red).

**Figure 3 ijms-24-04134-f003:**
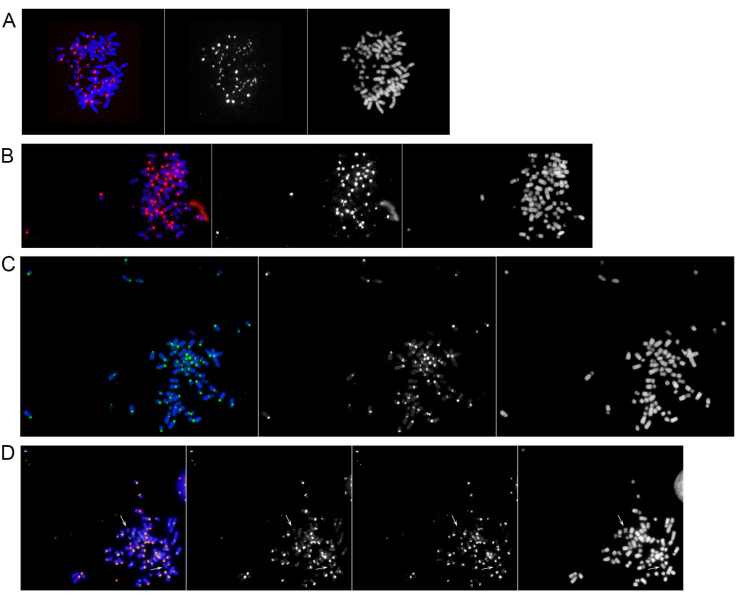
Hybridization of satellite DNA probes with metaphase chromosomes of *E. przewalskii*. Single-color FISH experiments with three satellite DNA probes: (**A**) 37cen (red signals), (**B**) 2PI (red signals), and (**C**) total genomic DNA (green signals). In panel (**D**), two-color FISH with 2PI probe (red signals) and total genomic DNA (green signals). The white arrows indicate chromosome 10, which is the only chromosome pair lacking any hybridization signal.

**Figure 4 ijms-24-04134-f004:**
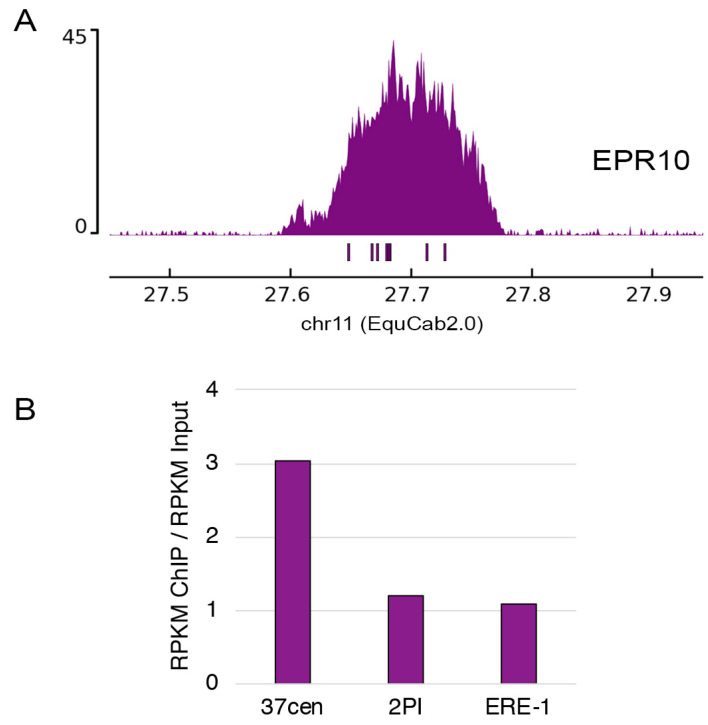
ChIP-seq characterization of centromeres in *Equus przewalskii*. (**A**) ChIP-seq identification of the EPR10 satellite-free centromere. ChIP-seq reads from primary fibroblasts of Przewalski’s horse were mapped on the EquCab2.0 horse reference genome. The CENP-A enriched domain is visualized as a peak. The *y*-axis reports the normalized read counts whereas the *x*-axis reports the coordinates on the reference genome. Under the peak, heterozygous nucleotide positions detected in the reads from immunoprecipitated DNA are indicated as rectangles. (**B**) Enrichment of 37cen and 2PI satellite families in the CENP-A immunoprecipitated chromatin. Values were measured as the ratio between normalized read counts (RPKM) in immunoprecipitated and input DNA. The ERE-1 retrotransposon is used as a control.

**Table 1 ijms-24-04134-t001:** Distribution of satellite DNA families on *Equus przewalskii* chromosomes.

EPR Chromosome	37cen		2PI		Genomic DNA	
	Number of Labeled Chromosomes (%)	Number of Analyzed Chromosomes	Number of Labeled Chromosomes (%)	Number of Analyzed Chromosomes	Number of Labeled Chromosomes (%)	Number of Analyzed Chromosomes
1cen	60 (100)	60	35 (85.4)	41	10 (100)	10
2cen	3 (5.1)	59	37 (100)	37	10 (100)	10
3cen	53 (100)	53	34 (100)	34	10 (100)	10
4cen	57 (100)	57	27 (71.1)	38	10 (100)	10
5cen	51 (100)	51	36 (100)	36	10 (100)	10
6cen	12 (20.3)	59	36 (100)	36	1 (10)	10
7cen	52 (100)	52	38 (100)	38	10 (100)	10
8cen	56 (100)	56	38 (100)	38	10 (100)	10
9cen	49 (100)	49	39 (100)	39	9 (100)	9
10cen	0 (0)	58	0 (0)	34	0 (0)	10
11cen	48 (100)	48	11 (40.7)	27	13 (100)	13
12cen	48 (100)	48	29 (100)	29	9 (100)	9
13cen	47 (100)	47	34 (100)	34	9 (100)	9
14cen	41 (100)	41	33 (100)	33	8 (100)	8
15cen	45 (100)	45	32 (100)	32	9 (100)	9
16cen	33 (100)	33	28 (100)	28	8 (100)	8
17cen	30 (100)	30	27 (100)	27	8 (100)	8
18cen	17 (44.7)	38	28 (100)	28	8 (100)	8
19cen	45 (100)	45	25 (100)	25	10 (100)	10
20cen	24 (100)	24	23 (100)	23	8 (100)	8
21cen	24 (100)	24	25 (100)	25	8 (100)	8
22cen	20 (100)	20	24 (100)	24	8 (100)	8
23cen	20 (100)	20	24 (100)	24	8 (100)	8
24cen	20 (100)	20	24 (100)	24	8 (100)	8
25cen	20 (100)	20	24 (100)	24	8 (100)	8
26cen	18 (100)	18	24 (100)	24	8 (100)	8
27cen	18 (100)	18	24 (100)	24	8 (100)	8
28cen	18 (100)	18	23 (100)	23	7 (100)	7
29cen	4 (10.5)	38	8 (36.4)	22	11 (100)	11
30cen	18 (100)	18	24 (100)	24	8 (100)	8
31cen	17 (100)	17	24 (100)	24	8 (100)	8
32cen	17 (100)	17	24 (100)	24	8 (100)	8
Xcen	29 (100)	29	15 (71.4)	21	5 (100)	5
Xqinter	0 (0)	29	0 (0)	21	5 (100)	5
Ycen	17 (100)	17	3 (25)	12	6 (100)	6

**Table 2 ijms-24-04134-t002:** SNV analysis of the genomic region corresponding to the CENP-A binding domain of chromosome EPR10.

Position (chr11, EquCab2.0 Reference)	Reference Allele	Alternative Allele	Number of Reads Reference Allele	Number of Reads Alternative Allele	Total Number of Reads	Reference Allele Frequency	Alternative Allele Frequency
27648739	T	C	8	15	23	0.3	0.7
27667251	T	C	20	15	35	0.6	0.4
27671967	C	G	14	18	32	0.4	0.6
27680741	G	C	7	9	16	0.4	0.6
27680744	C	T	8	8	16	0.5	0.5
27680746	C	T	9	9	18	0.5	0.5
27712271	C	T	11	19	30	0.4	0.6
27728102	T	A	8	16	24	0.3	0.7

## Data Availability

Raw sequencing data from this study have been submitted to the NCBI BioProject database (https://www.ncbi.nlm.nih.gov/bioproject/ (accessed on 1 February 2023)) under accession number PRJNA922937. The assembled mitochondrial DNA sequence of Przewalski’s horse from this has been submitted to the NCBI Nucleotide database (https://www.ncbi.nlm.nih.gov/nucleotide/ (accessed on 1 February 2023)) under accession number OQ285861.

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
