# Peer review of "A Satellite-Free Centromere in Equus przewalskii Chromosome 10"

_ijms, 2023, doi:10.3390/ijms24044134_

Round 1

Reviewer 1 Report

Tha article is well written but I have only small suggestions and questions. 

- I would recommend a reevaluation of the number of citations and where possible using more recent ones.

-I would also suggest a review of the title of the work that could better explain what is presented in the article.

- The authors referred in the paragraph 4.5(SNV analysis) "filtered for quality higher than 386 20. The minimum number of reads per position required for the analysis was 4"; are these values chosen on the basis of previous work or was it an arbitrary choice by the authors? Furthermore, Did the authors carry out a characterization of the cell line used to ensure that fibroblasts have been extracted?

Author Response

We would like to thank the Reviewer for the useful suggestions that allowed us to improve the manuscript.

Listed below are our answers to all specific points.

COMMENT: The article is well written but I have only small suggestions and questions. I would recommend a reevaluation of the number of citations and where possible using more recent ones.

ANSWER: We added new citations on Przewalski’s horse: Orlando et al. Bioessays 2020 (citation number 37), Bernatkova el al. BMC Zoology 2022 (citation number 44) and Jansen et al. PNAS 2002 (citation number 46).

COMMENT: I would also suggest a review of the title of the work that could better explain what is presented in the article.

ANSWER: We changed the title from “Centromeres and satellite DNA in Equus przewalskii” to “A satellite-free centromere in Equus przewalskii chromosome 10”.

COMMENT: The authors referred in the paragraph 4.5(SNV analysis) "filtered for quality higher than 386 20. The minimum number of reads per position required for the analysis was 4"; are these values chosen on the basis of previous work or was it an arbitrary choice by the authors?

ANSWER: We filtered for quality higher than 20 to exclude low quality calls, as suggested by the program. The minimum number of reads per position required for the analysis was 4, an arbitrary threshold that we previously used to identify SNVs in E. asinus (Nergadze et al. Genome Research 2018). To clarify this point, we modified the text in the Material and Methods section (lines 412-414).

COMMENT: Furthermore, Did the authors carry out a characterization of the cell line used to ensure that fibroblasts have been extracted?

ANSWER: The primary fibroblast cell lines were obtained from skin biopsies using a protocol well-established in our laboratory. Using this procedure, relatively pure cultures of skin fibroblasts can be obtained. The fibroblasts show the typical elongated shapes and grow aligned and in bundles when confluent. As shown also by other groups (Vangipuram et al. J Vis Exp 2013), the medium used supports the growth of fibroblasts whereas other cell populations such as keratinocytes need additional supplements and growth factors or are mitotically less active. Using this technique, our laboratory has derived over 30 fibroblast lines successfully. These details have been added to Materials and Methods section (lines 339-350). We also added references to our previous publications where fibroblast cell lines obtained with this protocol were used.

Reviewer 2 Report

The study provided information on two species known to be closely related and that the event of centromere repositioning which gave rise to centromeres found in the common ancestor, before the separation of the two equine lineages. The scientific merit of the work is very good.

From L 256……The authors claimed to have found a good degree of conservation of satellite DNA in Przewalski’s and in the domestic horse supported by a previous study.  Discussions in L 256 – 300: If necessary, the cited literature needs to be improved by reporting additional studies to support the claim (s).

The methods described in the study are sufficient. I could not find queries on the methodology that could disqualify the MS from being considered for publication in ijms. All results have been presented well and the conclusion provided by the authors is supported by the results.

The manuscript in general could be of more interest to researchers in this field.

Author Response

We would like to thank the Reviewer for the useful suggestions that allowed us to improve the manuscript.

COMMENT: The study provided information on two species known to be closely related and that the event of centromere repositioning which gave rise to centromeres found in the common ancestor, before the separation of the two equine lineages. The scientific merit of the work is very good.

From L 256……The authors claimed to have found a good degree of conservation of satellite DNA in Przewalski’s and in the domestic horse supported by a previous study.  Discussions in L 256 – 300: If necessary, the cited literature needs to be improved by reporting additional studies to support the claim (s).

The methods described in the study are sufficient. I could not find queries on the methodology that could disqualify the MS from being considered for publication in ijms. All results have been presented well and the conclusion provided by the authors is supported by the results.

The manuscript in general could be of more interest to researchers in this field.

ANSWER: In the Introduction, we added a sentence citing early studies on horse satellite repeats (lines 50-52) and added new references (reference numbers 24-27). In the Discussion section, we added description of the pioneering work from Ryder and Hansen on E. przewalskii satellite repeats that supports our conclusion (lines 302-306) (reference number 48).